# Dietary Diversity and Micronutrients Adequacy in Women of Childbearing Age: Results from ELANS Study

**DOI:** 10.3390/nu12071994

**Published:** 2020-07-04

**Authors:** Georgina Gómez, Ágatha Nogueira Previdelli, Regina Mara Fisberg, Irina Kovalskys, Mauro Fisberg, Marianella Herrera-Cuenca, Lilia Yadira Cortés Sanabria, Martha Cecilia Yépez García, Attilio Rigotti, María Reyna Liria-Domínguez, Viviana Guajardo, Dayana Quesada, Ana Gabriela Murillo, Juan Carlos Brenes

**Affiliations:** 1Departamento de Bioquimica, Escuela de Medicina, Universidad de Costa Rica, San Jose 94088, Costa Rica; dayana.quesada@ucr.ac.cr (D.Q.); anagrabriela.murillo@ucr.ac.cr (A.G.M.); 2Faculdade de Ciencias Biologicas e da Saude, Universidade Sao Judas Tadeu, Sao Paulo 01000, Brazil; agatha.previdelli@saojudas.br; 3Departamento de Nutriçao, Faculdade de Saude Publica, Universidade de Sao Paulo, Sao Paulo 01000, Brazil; rfisberg@usp.br; 4Committee of Nutrition and Wellbeing, International Life Science Institute (ILSI-Argentina), Buenos Aires C1059ABF, Argentina; ikovalskys@ilsi.org.ar (I.K.); viviana.guajardo@comunidad.ub.edu.ar (V.G.); 5Pontificia Universidad Catolica de Argentina, Facultad de Medicina, Buenos Aires B1675, Argentina; 6Instituto Pensi, Fundaçao Jose Egydio Setubal, Sabara Hospital Infantil, Sao Paulo 01239-040, Brazil; mfisberg.dped@epm.br; 7Departamento de Pediatria, Escola Paulista de Medicina, Universidade Federal de Sao Paulo, Sao Paulo 04023-062, Brazil; 8Centro de Estudios del Desarrollo, Universidad Central de Venezuela, Caracas 1010, Venezuela; marianella.herrera@ucv.ve; 9Departamento de Nutricion y Bioquimica, Pontificia Universidad Javeriana, Bogota 110111, Colombia; ycortes@javeriana.edu.co; 10Colegio de Ciencias de la Salud, Universidad San Francisco de Quito, Quito 17-1200-841, Ecuador; myepez@usfq.edu.ec; 11Centro de Nutricion Molecular y Enfermedades Cronicas, Departamento de Nutricion, Diabetes y Metabolismo, Escuela de Medicina, Pontificia Universidad Catolica, Santiago 833-0024, Chile; arigotti@med.puc.cl; 12Instituto de Investigacion Nutricional, La Molina, Lima 15026, Peru; rliria@iin.sld.pe; 13Instituto de Investigaciones Psicologicas & Centro de Investigacion en Neurociencias, Universidad de Costa Rica, San Jose 11501, Costa Rica; juan.brenessaenz@ucr.ac.cr

**Keywords:** Dietary diversity, nutrient adequacy, food groups, micronutrients, women of reproductive age

## Abstract

Dietary diversity, an important component of diet quality, is associated with an increased probability of adequate micronutrient intake. Women of childbearing age (WCA) are particularly vulnerable to micronutrient inadequacy. The Minimum Dietary Diversity for Women (MDD-W) has been used widely as a proxy measurement of micronutrient adequacy. This study aimed to assess the association between MDD-W and nutrients adequacy among WCA of eight Latin American countries. Nutrient intakes from 3704 WCA were analyzed with two 24-hour dietary recalls. Dietary diversity was calculated based on ten food groups with a cut-off point of intake ≥5 groups. The mean dietary diversity score was 4.72 points, and 57.7% of WCA achieved MDD-W. Vitamin D and E showed a mean Nutrient Adequacy Ratio (NAR) of 0.03 and 0.38, respectively. WCA with a diverse diet (MDD-W > 5) reported a significantly higher intake of most micronutrients and healthy food groups with less consumption of red and processed meats and sugar-sweetened beverages. MDD-W was significantly associated with the mean adequacy ratio (MAR) of 18 micronutrients evaluated. Nevertheless, even those women with a diverse diet fell short of meeting the Estimated Average Requirements (EAR) for vitamins D and E. MDD-W is an appropriate tool to evaluate micronutrients adequacy in WCA from Latin America, showing that women who achieved the MDD-W reported higher adequacy ratios for most micronutrients and an overall healthier diet.

## 1. Introduction

Latin American and Caribbean populations have experienced important epidemiologic, health, and nutritional transitions, marked by a growing tendency towards overweight and obesity, while still dealing with micronutrient deficiency and undernutrition [1]. Women of childbearing age (WCA) are a particularly nutritionally vulnerable population due to their higher physiological demands mainly related to their reproductive roles, such as an increased need for nutrients during menstruation, pregnancy, and lactation [2]. Additionally, social and economic disadvantages may further exacerbate this vulnerability [3].

The consumption of a varied and balanced diet during this critical age window is essential, as a woman’s current and future wellbeing may be affected by nutrient inadequacy in terms of increased susceptibility to diseases and impaired growth, development, and productivity. Moreover, micronutrient deficiencies can adversely influence fertility, pregnancy outcomes, and risk of congenital disabilities, compromising both the mother and offspring’s health [4]. Studies describing micronutrient intake in WCA of Latin American countries within representative samples of the population are scarce.

To respond to the Sustainable Development Goal (SDG) proposed by the United Nations in 2015 [5], monitoring the nutritional status and dietary intake of populations is imperative. Tracking dietary diversity and dietary quality could guide nutritional interventions that help ensure nutritional security and sustainable food production. In this context, dietary intake assessments that provide detailed quantitative data are not always affordable for many low- and middle-income countries (LMIC), which increases the need for a feasible and straightforward indicator of diet quality [6].

The Dietary Diversity Score (DDS) is currently used as an indicator of micronutrient adequacy [7]. Since a single food cannot provide all necessary nutrients for optimal health, the consumption of an appropriate combination of various foods helps to ensure nutrient adequacy. DDS quantifies the number of food items or food groups consumed over a reference period, can be measured in the household or at an individual level, and has long been recognized as a critical element of diet quality [8]. A diverse diet has been associated with an increased consumption of shortfall nutrients (i.e., vitamin A, vitamin D, vitamin E, folate, calcium, iron, and magnesium) in WCA, improving their nutritional and health parameters [6,9,10].

The Minimum Dietary Diversity for Women (MDD-W) of reproductive age developed by the Food and Agricultural Organization (FAO) of the United Nations in 2016 is a proposal of a single indicator to assess dietary quality in women of reproductive age. According to this methodology, women who achieve the minimal diet diversity, i.e., consuming five or more food groups, are expected to have a higher likelihood of meeting their micronutrient intake recommendations compared to those who consume fewer food groups [11]. The MDD-W has been widely used to compare the dietary diversity of female populations across different contexts [12,13,14]. Measured on an individual level, MDD-W has been used as a proxy measurement for diet quality and micronutrient adequacy, showing associations with nutrient adequacy [10,13,15,16]. Alternatively, a diverse diet may also be associated with more high-energy food sources and nutrients that represent a public health concern, such as added sugars, saturated fat, and sodium. Therefore, a diverse diet could also lead to unhealthy weight gain and chronic non-communicable diseases in adults [17,18]. The latter suggests that within a diverse diet, it is also important to assess diet healthfulness, namely, adequate food consumption as defined by dietary guidelines. The MDD-W has been used in low- and middle-income countries (LMIC) of Asia and Africa; however, very few studies have used this tool in Latin American countries. This study aimed to assess MDD-W in relation to micronutrient adequacy and healthier food intake among women of childbearing age of eight Latin American countries.

## 2. Methods

### 2.1. Study Population

Data for this analysis were obtained from the Latin American Study on Nutrition and Health/Estudio Latino Americano de Nutrición y Salud (ELANS), a household-based cross-sectional, multicenter survey that provides comparable data of dietary intake and physical activity, and their associations with anthropometric profiles among representative urban populations of eight Latin American countries. Data were collected from September 2014 to August 2015. Sample recruitment was performed through a random complex, multistage process stratified by geographical location (only urban areas), sex, age, and socioeconomic status (SES), with a sample error of 3.49% at a 5% statistical significance level. More information was described in detail elsewhere [19].

A total of 9218 subjects (4409 males and 4809 females) aged 15 to 65 years living in urban areas of Argentina (average age 31.8 ± 9.4), Brazil (31.9 ± 9.3), Colombia (31.2 ± 10.3), Costa Rica (30.7 ± 10.1), Chile (31.1 ± 9.9), Ecuador (30.7 ± 10.1), Peru (29.9 ± 9.2), and Venezuela (30.5 ± 10.2) were included in the study. For this analysis, only non-lactating, non-pregnant women of childbearing age (15–49 years old) were included (*n* = 3704) [20]. SES was evaluated using a country-specific questionnaire based on the legislative requirements or established local standard layouts and were classified as high, medium, and low status [19]. The ELANS protocol was approved by the Western Institutional Review Board (#20140605) and the Ethics Review Boards of each of the participating institutions and was registered at Clinical Trials (#NCT02226627). Written informed consent was obtained from all study participants. Individual confidentiality for the pooled data was maintained by using numeric identification codes rather than names. All data transfer was done with a secure file sharing system.

### 2.2. Anthropometric Measurements

Anthropometric measurements were obtained from all participants by trained interviewers following standardized procedures [21]. Body weight was measured after all heavy clothes, pocket items, shoes, and socks were removed, using a calibrated electronic scale up to 200 kg with an accuracy of 0.1 kg. Height was measured with a portable stadiometer up to 205 cm with an accuracy of 0.1 cm. Waist, hips, and neck circumferences were measured with an inelastic tape to the nearest 0.1 cm. Body mass index (BMI; weight (kg)/height (m^2^)) for participants under 18 years old was classified according to the percentile or z-score cut-off criterion for age and sex proposed by the World Health Organization (WHO) [22]; for those over 18 years old, BMI was defined following the WHO BMI classification: underweight if BMI was ≤18.5 kg/m^2^, normal weight if BMI > 18.5–24.99 kg/m^2^, overweight if BMI ≥ 25.0–29.9 kg/m^2^, and obese if BMI ≥ 30.0 kg/m^2^ [23].

### 2.3. Dietary Assessment

Dietary intake was collected by trained interviewers during two face-to-face household visits, using 24-hour recalls (24 h) in two non-consecutive days, with an interval up to eight days between them, including both weekdays and weekend days, with a proportional distribution of days among the sample, in order to capture day-to-day variation in food consumption. To assess all foods and beverages consumed over the previous day, a 24 h recall was conducted following the United States Department of Agriculture (USDA) five-step multiple-pass method [24]. A photographic album of common foods of each country and household utensils were used to estimate portion sizes. All local and traditional foods reported were harmonized with a USDA composition table considering the nutritional equivalency [25]. Collected data were converted into grams and milliliters, and energy, macronutrients, and micronutrients quantities were obtained using the Nutrition Data System for Research (NDS-R) software version 2014, developed by the Nutrition Coordinating Center of the University of Minnesota, Minneapolis.

As proposed by the European Prospective Investigation into Cancer and Nutrition (EPIC), the Multiple Source Method (http://mss.dife.de/tps/en) was used to estimate usual intake of energy, carbohydrates, proteins, fats (total, saturated, monounsaturated, polyunsaturated, trans fats, and cholesterol), minerals (calcium, iron, sodium, phosphorous, magnesium, zinc, and selenium), and vitamins (thiamin, riboflavin, niacin, pyridoxine, cobalamin, vitamin A (as retinol equivalents), folate equivalents, vitamin C, vitamin D, and vitamin E). Dietary intakes were adjusted to 1000 kcal per day to allow comparisons among women with a diverse diet and those with a non-diverse diet, independently of diet quantity, and to reduce measurement error due to energy intake under- or over-reporting.

### 2.4. Dietary Diversity Score

Dietary Diversity Score (DDS) was assessed at an individual level. Since the FAO protocol to measure dietary diversity is intended to be used in large-scale surveys as a simple data collection approach, food consumption is assessed applying one 24 h recall; therefore, for the purpose of this study, only the first 24 h recall was used to calculate DDS. All food items reported to be consumed during the first 24 h recall were classified into ten food groups, according to the MDD-W [11]: (1) starchy staples (grains, with roots and tubers, and plantains); (2) meat, poultry, and fish; (3) dark green leafy vegetables; (4) other vitamin A-rich fruits and vegetables; (5) other vegetables; (6) other fruits; (7) pulses (beans, peas, and lentils); (8) dairy; (9) eggs; and (10) nuts and seeds (Table 1). To the consumption of at least 15 g/day of each food group was assigned 1 point (if consumed) or 0 points (if intake of that specific food group was less than 15 g/day). For each individual, a minimum of 0 and a maximum of 10 points could be obtained. Higher scores indicate higher diversity, as more food groups were reported to be eaten. To achieve minimal dietary diversity, respondents must consume foods from at least five of the ten food groups. DDS analysis was performed by country, age group, SES, and nutritional status.

### 2.5. Assessment of Nutrient Adequacy

To estimate the nutrient adequacy of the diet, the nutrient adequacy ratio (NAR) was calculated for 17 out of the 18 micronutrients assessed: calcium, iron, vitamin A, vitamin C, vitamin D, vitamin E, thiamin, riboflavin, niacin, cobalamin, pyridoxine, zinc, magnesium, copper, folate, phosphorous, and selenium, but not for sodium. Although sodium is an essential nutrient, there is insufficient scientific evidence of a causal relationship between intake of sodium and an indicator of adequacy, as well as, evidence of an intake–response relationship for this nutrient to establish an Estimated Average Requirement (EAR) [26]. The NAR value for a given nutrient is the ratio of a respondent’s current intake of the nutrient to the EAR for the corresponding age category. A NAR = 1 indicates a value that is 100% of EAR, meaning that the intake equals the requirement. EAR values were used because they are recommended as the standard parameters to estimate the prevalence of inadequate nutrient intake within a group [27]. The mean adequacy ratio (MAR) was calculated as the sum of all NARs divided by the number of nutrients assessed (*n* = 17). NARs were truncated at 1 so that a nutrient with a high NAR cannot compensate for a nutrient with a low NAR. An adequacy ratio of 0.6 was used as a cut-off point for nutrient adequacy to ensure comparability with previous multi-country analyses [6,10]. MARs were compared by age group, country, SES, nutritional status, and dietary diversity accomplishment.

### 2.6. Statistical Analyses

Data were analyzed using the Statistical Package for Social Sciences (SPSS) software program (version 23, SPSS Inc., Chicago, IL, USA). Data were reported as mean ± standard deviations (SD) for continuous variables and as frequencies (i.e., percentages) for categorical variables. The between-group comparisons were analyzed with factorial variance analysis (ANOVA) followed by Fisher-protected Lowest Statistical Difference (LSD) post-hoc test, when appropriate. Eta squared coefficients (η^2^) were estimated as an index of the effect size and were expressed in percentages. A student *t*-test for one sample was used to compare the whole sample to the cut-off criterion for a diverse diet (>5). A chi-square test was used to estimate the significant differences in the distribution of participants (diverse versus non-diverse diet) for each of the food groups evaluated (if eaten or not). Pearson’s correlations coefficients were used for determining the association between DDS and NAR/MAR scores. Stepwise, multiple linear regression models were built to determine the best NAR predictors to the DDS and MAR scores. Partial correlation analyses were employed to adjust and control Pearson and regression coefficients for sociodemographic, nutritional, and anthropometric variables, when appropriate. A binomial logistic regression analysis was performed to estimate the odds ratios for belonging to the diverse (1) versus the non-diverse diet subgroup (0), using all NAR micronutrients as predictors. The final model was obtained by including and removing on successive blocks the variables according to their respective eta squared coefficients until no further significant contributions to the whole model were added. These analyses were computed with the statistical package Jamovi (Jamovi project 2018, Version 1.2.9, Sydney, Australia), retrieved from https://www.jamovi.org). *p* < 0.05 was considered as statistically significant.

## 3. Results

### 3.1. Diet Diversity Score (DDS) According to Sociodemographic Variables

Table 2 shows the Diet Diversity Score (DDS) according to the sociodemographic variables and nutritional status of ELANS. The mean DDS for the whole sample was 4.73 ± 1.34 out of 10 possible points maximal score. This mean value was lower (*t*
_(3703)_ = −12.479, *p* < 0.0001) than the recommended cut-off criterion (five or more food groups consumed) for a diverse diet [11]. Out of the total sample, 57.7% of the participants could be classified as having a diverse diet based on the cut-off criterion (Table 2). However, none of the respondents consumed foods from all groups examined (Table 1). There was a main effect of country (*F*
_(7,3696)_ = 30.207, *p* < 0.0001, η^2^ = 0.054), in which only Peru and Ecuador had average scores above five points. Also, the DDS varied between SES (*F*
_(2,3701)_ = 17.696, *p* < 0.0001, η^2^ = 0.009), with people of low SES having significantly lower scores than those in the high SES (LSD, *p* < 0.05), which did not differ from the medium SES. No main effects for the nutritional status and age were observed. However, an interaction between country, SES, and nutritional status (*F*
_(39,3608)_ = 1.462, *p* < 0.032, η^2^ = 0.016) revealed that the DDS was higher in underweighted and normal-weight women from the high SES in Peru and Ecuador. When analyzing the sociodemographic variables in this subsample, the main effects of country (*F*
_(7,2130)_ = 12.770, *p* < 0.0001, η^2^ = 0.04) and SES were retained (*F*
_(2,2135)_ = 3.591, *p* < 0.028, η^2^ = 0.003; Table 2).

### 3.2. Consumption of Food Groups

The respondents consumed foods from a range of 1–9 groups with 42.3% of the sample (1566) consuming from 1 to 4 food groups (i.e., non-diverse diet), 30.4% (1127) consuming from five groups (i.e., acceptable diverse diet), and 27.3% (1011) consuming more than five groups (i.e., highly diverse diet). Figure 1 shows the percentage of participants with a diverse and non-diverse diet according to the food groups analyzed (Table 1), which were ranked based on their preference in the study sample. The food groups that were consumed by more than 50% of the participants were starchy staples (99.4%), meat (84.2%), other vegetables (71.7%), and dairy products (71.0%). Less than 50% of the participants reported intake of fruits (41.6%), eggs (35.6%), pulses (31.2%), and vitamin A-rich vegetables and fruits (28.2%). The lowest consumption was for green leafy vegetables (6.8%) and nuts and seeds (2.8%).

When comparing the percentage of women from the diverse and non-diverse subgroups regarding the consumption of each food group, there were significantly more subjects in the diverse diet subgroup consuming those foods. The largest between-group differences in the percentage of participants were observed in the following order: dark green leafy vegetables (Δ = 88%; χ*^2^*_(1,253)_ = 196.56, *p* < 0.0001); nuts (Δ = 81%; χ*^2^*
_(1,105)_ = 81, *p* < 0.0001); eggs (Δ = 52%; χ*^2^*
_(1,1320)_ = 360.68, *p* < 0.0001); pulses (Δ = 46%; χ*^2^*
_(1,1156)_ = 254.12, *p* < 0.0001); other vegetables (Δ = 43%; χ*^2^*
_(1,2654)_ = 493.12, *p* < 0.0001); meat, poultry, and fish group (Δ = 36%; χ*^2^*
_(1,3119)_ = 198.58, *p* < 0.0001); dairy (Δ = 34%; χ*^2^*
_(1,2631)_ = 308.55, *p* < 0.0001); fruits (Δ = 20%; χ*^2^*
_(1,1541)_ = 589.36, *p* < 0.0001); starchy staples (Δ = 16%; χ*^2^*
_(1,3680)_ = 93.31, *p* < 0.0001); and vitamin A-rich fruits and vegetables (Δ = 14%; χ*^2^*
_(1,1045)_ = 577.73, *p* < 0.0001).

### 3.3. Energy, Nutrients, and Food Groups Intake in Diet Diversity Subgroups

We compared energy, macronutrient, micronutrient, and food group consumption between participants with a non-diverse (DDS < 5) versus a diverse diet (DDS ≥ 5; Table 3). Out of the ten macronutrients evaluated, six were significantly different between groups, with the omega-3 fatty acids as the most important differentiating factor of a diverse diet with an explained variance of 4% (Table 3). In the second place was energy intake (3%), with a higher mean value in the diverse diet group. In third place appeared added sugars, which were higher in the non-diverse group with an explained variance of 2%. Trans fatty acids, cholesterol, and monounsaturated fats were also significantly different between groups, but with rather negligible size effects (<1%). Out of the eighteen micronutrients evaluated, thirteen were significantly different between groups, with vitamin A (3%), magnesium (3%), pyridoxine (2%), vitamin D (2%), and phosphorous (2%) as the micronutrients with higher intake by a highly diverse diet. The consumption of food groups yielded significant between-group differences for all of them, except for fish. In fact, from all variables analyzed, the largest size effects were obtained for fruits (5%), fiber (4%), and vegetables (2%) as the most defining foods of a diverse diet. Those with a non-diverse diet reported significantly higher consumption of processed red meat (2%) and sugar-sweetened beverages (1%). None of these comparisons were lost after controlling by country, age, and SES, indicating that significant differences were not affected by other confounding variables.

### 3.4. Nutritional Status and Anthropometric Measurements in Diet Diversity Subgroups

When comparing anthropometric measurements between participants with a non-diverse (DDS < 5) and a diverse diet (DDS ≥ 5; Table 4), no significant differences were observed for any of the variables analyzed, even after controlling by country, age, and SES.

### 3.5. Nutrient Adequacy Ratio (NAR)

Out of the 17 nutrients assessed, vitamin E and vitamin D, showed an adequacy ratio below 70% of EAR (NAR < 0.7) in all countries, with an overall mean NAR for vitamin E of 0.031 ± 0.02 (ranging from 0.019 in Brazil and 0.020 in Venezuela to 0.051 in Colombia) and an overall mean NAR of vitamin D of 0.343 ± 0.21 (ranging from 0.192 in Brazil to 0.564 in Ecuador). Another shortfall micronutrient observed was calcium with an overall mean NAR of 0.634 ± 0.46, showing NARs < 0.7 in Costa Rica (0.417 ± 0.49), Brazil (0.449 ± 0.49), Peru (0.545 ± 0.49), and Chile (0.553 ± 0.49). Folate and magnesium were also identified as shortfall nutrients in some, but not in all assessed countries. The mean NAR of folate was 0.702 ± 0.18, and was <0.7 in Chile (0.649 ± 0.18), Colombia (0.652 ± 0.15), Venezuela (0.663 ± 0.16), Peru (0.664 ± 0.16), and Costa Rica (0.665 ± 0.17), while the magnesium mean NAR was <0.7 in Chile (0.648 ± 0.15; Appendix A).

Table 5 shows the NAR values comparing participants with a non-diverse diet (DDS < 5) and a diverse diet (DDS ≥ 5). Out of the 17 micronutrients assessed, only selenium was not significantly different between the groups. The NAR values were higher in the high diverse diet subgroup, except for folate, which was higher in the non-diverse diet group. A further analysis revealed that synthetic folic acid intake was higher among women with DDS < 5, probably because of the higher consumption of fortified cereals. As most of the NARs showed the same trend between groups, the MAR values were, in consequence, significantly higher in the diverse diet group. The largest effect sizes were obtained for magnesium, vitamin A, and vitamin C (all with 9%), followed by pyridoxine (5%) and vitamin D (4%). All group differences for NAR and MAR scores remained the same after controlling by country, age, and SES. Pearson correlation coefficients were determined for each NAR micronutrient in relation to the DDS for the whole sample. All the NAR micronutrients correlated positively and significantly with the DDS, except for folate, which correlated negatively with the DDS (Table 5). Higher correlation coefficients were obtained, as expected, for magnesium, vitamins A, C, and D, and pyridoxine. When the micronutrient NAR with higher correlation coefficients were examined for competition among each other, the stepwise multiple linear regression model placed magnesium as the best predictor (*R^2^* = 0.044, *p* < 0.0001) of DDS, with vitamin A (2.2%) and vitamin D (0.8%) adding minor yet significant contributions to the overall prediction. Also, the MAR score correlated positively with the DDS (*r* = 0.393, *p* < 0.0001). Such an association remained almost the same after controlling for energy intake (*r* = 0.338, *p* < 0.0001), body weight (*r* = 0.392, *p* < 0.0001), country (*r* = 0.392, *p* < 0.0001), age (*r* = 0.393, *p* < 0.0001), and SES (*r* = 0.397, *p* < 0.0001), indicating that despite being moderate, the relationship between MAR and DDS was rather consistent.

Subsequently, we performed a stepwise multiple linear regression to determine which combination of micronutrient NARs contributed the most to the MAR score, as NAR and MAR values are often recommended to estimate the prevalence of inadequate dietary intake within a given population. The most important micronutrient NAR contributing to the MAR score was vitamin D (*R^2^* = 0.17, *p* < 0.0001), followed by the linear combination of vitamin D + calcium (*R^2^* = 0.25, *p* < 0.0001). The other micronutrients with significant coefficients made rather small contributions (<0.5%) to the MAR score. In the case of the food groups, the consumption of dairy was the best predictor of MAR scores (*R^2^* = 0.04, *p* < 0.0001), followed by the combination of dairy + beans (*R^2^* = 0.08, *p* < 0.0001) and dairy + beans + fiber (*R^2^* = 0.10, *p* < 0.0001). From there on, the other food groups with significant coefficients (e.g., nuts, vegetables, and fruits) made negligible contributions to the overall prediction of MAR scores, with changes in the *R^2^* coefficients ranging from 0.6% to 0.4%. It is worth noting that those significant predictions remained almost the same or even improved after controlling for age, SES, BMI, country, and energy intake, indicating that the predictions are rather stable despite being relatively small.

Finally, we performed a binomial logistic regression analysis to estimate the odds ratios for belonging to the diverse diet group. From all micronutrient NARs, the analysis retained four variables with odds ratios ranging from 7.31 to 2.34 for vitamin A and vitamin D, respectively (Table 6). After controlling for age, country, and SES, all odd ratios increased with vitamin C showing a slightly higher odds ratio than vitamin D (Table 6). Although magnesium, vitamin A and vitamin C had the same eta squared coefficients (Table 5), vitamin A was a better predictor of a diverse diet together with vitamin C. Vitamin D had a lower eta squared coefficient than magnesium. Although pyridoxine had a high eta squared coefficient when combined with other predictors in the previous linear regression models, it was not retained in the binomial logistic regression model when placed to compete with the other variables.

## 4. Discussion

The present study, conducted among women of the reproductive age of urban populations from eight Latin-American countries, provides evidence that MDD-W is a good proxy for most micronutrients assessed. MDD-W was associated not only with a higher intake and NAR of most of the assessed micronutrients, but also with greater consumption of healthy food groups, and less consumption of red and processed meat and sugar-sweetened beverages. 

DDS showed an average score lower than the 5-point cut-off proposed by FAO [11]. When analyzed by country, only Peru and Ecuador reached the MDD-W. Several studies conducted in African populations using the same methodology have obtained worse results [7,28,29,30]. According to the MDD-W threshold of five or more food groups, 57.7% of WRA living in Latin America have a diverse diet. Peru and Ecuador had the highest percentage of women with a diverse, whereas Argentina had the lowest percentage. Our results showed to be better than the 25% of WRA reaching a diverse diet in Gitagia, Kenya (2019) [30], and Chakona, South Africa (2017) [29]. In the study of Bellows and colleagues (2019), only 10% of women of reproductive age in rural Tanzania consume at least five food groups [28], far below what was seen in our findings. Also, in our study, women with low SES had significantly lower DDS than those in the medium and high SES (*p* < 0.05). This pattern has been reported previously [17,31,32,33,34], suggesting that women with higher purchasing power have access to a wider variety of food sources leading to better diet quality. However, other studies have shown no correlation between SES and dietary diversity [35].

In the present study, starchy foods were the food group reported by nearly all the population probably due to its low cost and high caloric density. These foods are more resourceful in terms of satisfying family meals at a cheaper price compared with protein sources and vegetables that are more expensive and difficult to access for the low-income population. Similar findings have been reported in Honduran, Sri Lankan adults, and other populations [33,36,37,38]. In agreement with our results, the food groups less reported by the Honduran population were dark green leafy vegetables as well as nuts and seeds [38]. In terms of the number of persons eating food groups, there were more women who reached the criteria for dietary diversity consuming fruits and vegetables (including vitamin A-rich fruits and vegetables), eggs, and dairy. Regarding the amount consumed of these foods, the women with a diverse diet ate more fruits, fiber, and vegetables. Similar trends in consumption have been found in other world regions for fruits [17,30,33,39], vitamin A-rich foods [17,30,40], and non-starchy vegetables [41]. We found a higher energy intake in women with MDD-W-5, in agreement with previous studies [33,42], and higher consumption of omega-3 fatty acids from plants, which was one of the most important differentiating factors of a diverse diet in our study. In this subgroup, there was also a higher intake of almost all micronutrients assessed with larger differences being observed for vitamin A, magnesium, pyridoxine, vitamin D, and phosphorous. Women reaching the criteria for dietary diversity reported lower consumption of monounsaturated and trans fats, sodium, sugar-sweetened beverages, and processed and not processed red meats, which are recognized as cardiovascular disease, diet-related risk factors [41]. Previous research has found a decreased probability of diabetes, hypercholesterolemia, and hypertension with increasing consumption of whole grains, vegetables, and calcium-rich foods, respectively [42]. Studies have indeed suggested that there is a positive relation between fruit and vegetable intake and the overall diet quality [43,44]. Farhangi and Jahangiry (2018) found lower serum triglyceride and systolic blood pressure and higher serum adiponectin concentrations in top quartiles of dietary diversity score in patients with metabolic syndrome from Iran, establishing a positive association between healthy dietary parameters and cardiometabolic risk factors [45]. This finding contrasts with other population-based observational studies reporting no benefit of diet quality associated with increased food diversity [44,46], which might be attributed to cultural and methodological differences for assessing food consumption.

Despite a higher energy intake in women with a diverse diet, we found neither differences in nutritional status nor in anthropometric measurements when comparing the dietary diversity subgroups. Previous studies have reported higher DDS among obese than in normal BMI subjects [47,48] and a greater dissimilarity among foods associated with gaining waist circumference [36]. This inconsistency of our findings with previous studies may be due to the different methodologies used to evaluate this association, such as the use of different scoring methods or additional adjusting approaches for energy intake, age, and other confounding covariables.

In addition, we found a positive association between DDS and the chance of micronutrient adequacy, consistent with previous studies [13,16,43,49]. In that regard, the best NAR predictors of DDS were magnesium, vitamin A, and vitamin D. Also, the DDS appeared to be moderately associated with the mean adequacy ratio (MAR), which was significantly higher for those with a diverse diet. The most important NAR micronutrients contributing to the MAR score were vitamin D and calcium. Despite these findings, the MDD ≥ 5 cut-off did not perform well for vitamin D, vitamin E, and calcium, which showed mean NARs below 70% of EAR—even in the diverse diet subgroup. Not even those with DDS of nine points reached the cut-off point of 0.6 for nutrient adequacy for vitamin D or vitamin E. Our results clearly indicate that NAR, MAR, and DDS scores are quite consistent among each other showing theoretically sound associations with macro- and micronutrients representative of a diverse diet. Among all NARs studied, vitamins A, C, and D, and magnesium exhibited the highest odd ratios for belonging to the diverse diet group. However, the shortfall in the EAR intake for some of these micronutrients could undermine the extent and significance of our findings. On the other hand, it is worth noting that dietary assessment of some vitamin intake having a large day-to-day variation may require a food-frequency questionnaire that gauges more accurately the intake over longer periods. In the case of vitamin D, biosynthesis in the skin should also be considered.

The low dietary diversity in WCA of Latin American women was mainly due to cereal-based diets with low consumption of nutrient-rich foods, including fruits and vegetables. This macro- and micronutrient imbalance can impose a large burden on women’s health, leading to loss of productivity and increased risk of chronic diseases. In addition, an overall unhealthy diet and lifestyle before pregnancy in WCA has been associated with a higher risk of offspring’s obesity in childhood, adolescence, and early adulthood [50].

To our knowledge, this is the first study that assessed the relationship among the MDD-W proposed by FAO and nutrient adequacy in a multicenter study performed in the Latin American population. Given the cost and complexity of national food consumption surveys, LMIC has the need to identify simple indicators of diet quality and micronutrient adequacy to monitor food security and measure the impact of nutrition programs and public policies. Although the MDD-W is based in on a single 24 h recall and might not be representative of the overall food intake, the idea was to have a method that can be used in situations in which simplicity superimposes to accuracy in terms of quantitative assessment of food consumption. Findings from this study provide evidence that MDD-W is a good proxy of micronutrient intake in women of childbearing age from the Latin American population. However, we are aware that our study has some limitations. First, there is always a bias when collecting dietary intake data in general, and especially with single 24 h recalls. In addition, our data are limited to urban populations, thus, it does not represent the rural populations. Nevertheless, the use of a large, nationally representative sample of the urban population from Latin America is one of our main strengths, together with the acquisition of quantitative data by means of standardized methods with simultaneous data collection allowing for better are more reliable comparisons between countries [19].

## 5. Conclusions

This study revealed that dietary diversity is limited among Latin American countries. A higher DDS was associated with a healthier diet, in terms of a higher intake of micronutrients, greater consumption of healthy food groups, and lower intake of trans fatty acids, added sugar, and sodium. Nutritional interventions emphasizing the importance of maximizing dietary diversity and fruit and vegetable intake should be encouraged to ensure optimum nutritional adequacy in this world region. Public interventions are needed to promote an adequate diversity of the diet, with a special emphasis on the importance of fruit and vegetable consumption while controlling energy intake. Given the cost and complexity of national food consumption surveys, MDD-W becomes an important indicator for assessing micronutrient adequacy of the diet indirectly in population of low- and medium-income countries and to monitor and evaluate intervention programs and public health policies aimed to address the Sustainable Development Goals proposed in 2015 by the United Nations.

## Figures and Tables

**Figure 1 nutrients-12-01994-f001:**
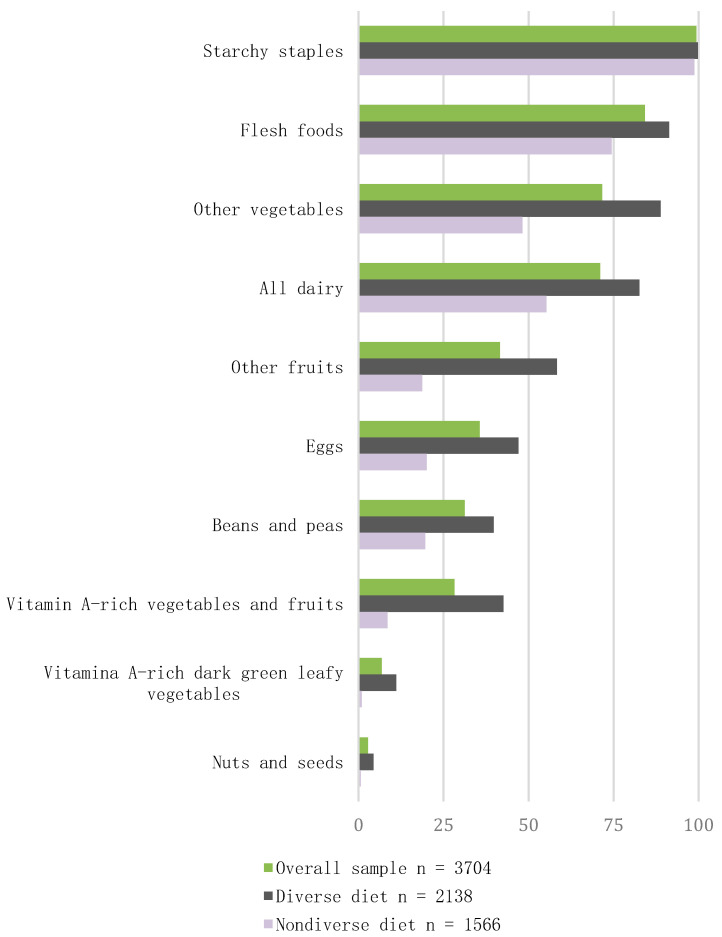
Proportion (%) of participants consuming each food group over one 24 h recall period in ELANS. Overall (*n* = 3704), diverse (*n* = 2138) consuming ≥5 food groups, and non-diverse (*n* = 1566) consuming <5 food groups.

**Table 1 nutrients-12-01994-t001:** Food groups included in the minimum Dietary Diversity Score.

Food Group	Specific Foods
Starchy staples (Grains, White Roots, Tuber, and Plantains)	Rice, Bread, Tortillas, Breakfast Cereals, Maize, Pasta, Cassava, Potatoes, Plantains Rip, and Green
Meat Poultry and Fish	Beef, Pork, Lamb, Veal, Chicken, Turkey, Liver, Other Organs, Canned Sardines and Tuna, Fresh Fish, and Seafood
Dark Green Leafy Vegetables	Broccoli, Mustard Greens, Turnip Greens, Collards, Spinach
Other Vitamin A-rich Fruits and Vegetables	Carrots, Sweet Potatoes, Winter Squash, Pumpkin, Cantaloupe, Mango, Papaya, and Apricot
Other Vegetables	Lettuce, Mixed Greens, Tomatoes, Cauliflower, Radish, Okra, Green Peas, Green Pepper, Onions, Shallots, Leeks, String Beans, and Others
Other Fruits	Oranges, Grapefruits, Banana, Apples, Pears, Strawberries, Watermelon, Kiwi, Berries, Melons, Avocado, Lemon, Lime, Tangerine, Pineapple, Tamarind, and Others
Pulses (Beans, Peas, and Lentils)	Lentils, Beans (Black, Kidney, Pinto, and Others), Chickpeas and Soybeans
Dairy	Milk, Yogurt, and Cheese
Eggs	Eggs
Nuts and Seeds	Pecans, Cashews, Peanuts, Almonds, Walnuts, Sunflower Seeds, other Seeds

**Table 2 nutrients-12-01994-t002:** Sample characteristics, Diet Diversity Score (DDS) and proportion of participants reaching the minimal DDS (consumption of at least five out of 10 Food Groups) in ELANS.

Dietary Diversity Score (DDS)					Participants Reaching the Minimum DDS				
	*n*	Mean	SD	*p*	η^2^ (%)	*n*	%	*p*	η^2^ (%)
Overall	3704	4.73	1.34			2138	57.7		
Age ranges									
15–19	539	4.61	1.28	0.081	0.1	288	13.5	0.365	0.1
20–34	1771	4.73	1.34			1019	47.7		
35–49	1394	4.76	1.35			831	38.9		
**Socioeconomic Status**									
High	529	4.96	1.35	0.001	0.9	339	64.1	0.028	0.3
Medium	1593	4.78	1.33			956	60.0		
Low	1582	4.59	1.33			843	53.3		
**Country**									
Argentina	521	4.35	1.35	0.001	5.4	236	45.3	0.001	4
Brazil	798	4.61	1.34			435	54.5		
Chile	345	4.71	1.23			205	59.4		
Colombia	464	4.71	1.36			270	58.2		
Costa Rica	309	4.90	1.36			190	61.5		
Ecuador	324	5.16	1.30			228	70.4		
Peru	480	5.28	1.29			347	72.3		
Venezuela	463	4.38	1.15			227	49.0		
Nutritional Status									
Underweight	128	4.81	1.33	0.252	0.1	77	60.2	0.764	0.1
Normal Weight	1444	4.73	1.31			834	57.8		
Overweight	1177	4.77	1.33			690	58.6		
Obesity	952	4.66	1.38			534	56.1		

SD: Standard deviation. η^2^: Eta squared coefficients for estimating the effect size.

**Table 3 nutrients-12-01994-t003:** Consumption of energy, nutrients, and food groups according to Dietary Diversity in ELANS.

Variables	DDS < 5		DDS ≥ 5		*p*	η^2^ (%)
	Mean	SD	Mean	SD		
Energy (kcal)	1721.75	527.00	1902.50	505.93	0.0001	3
Macronutrients *						
Omega-3 Fatty Acids from Plants (g)	0.04	0.03	0.06	0.03	0.001	4
Added Sugars (g)	36.86	16.12	32.53	13.10	0.001	2
Trans Fatty Acids (g)	1.28	0.88	1.19	0.82	0.001	0.3
Cholesterol (mg)	144.83	48.25	150.30	45.32	0.001	0.3
Monounsaturated Fats (g)	11.27	2.57	11.04	2.47	0.005	0.2
Protein (g)	40.20	7.88	40.70	7.37	0.050	0.1
Saturated Fats (g)	11.11	2.91	10.94	2.81	0.068	0.1
Total Fats (g)	33.95	6.46	33.64	6.16	0.137	0.1
Polyunsaturated Fats (g)	8.68	2.26	8.59	2.04	0.193	0.01
Carbohydrates (g)	138.10	20.94	138.67	20.13	0.403	0.01
Micronutrients *						
Vitamin A (mg)	293.56	151.00	345.36	149.46	0.001	3
Magnesium (mg)	117.72	25.73	126.84	25.75	0.001	3
Pyridoxine (mg)	0.85	0.23	0.90	0.21	0.001	2
Vitamin D (mg)	1.73	1.12	2.02	1.09	0.001	2
Phosphorous (mg)	524.29	111.01	552.35	102.24	0.001	2
Folate Equivalents (mg)	262.13	70.96	276.90	66.91	0.001	1
Zinc (mg)	6.31	3.28	5.80	2.18	0.001	0.8
Vitamin C (mg)	48.30	75.29	57.81	47.24	0.001	0.6
Calcium (mg)	290.82	115.83	307.45	106.89	0.001	0.5
Cooper (mg)	0.82	0.76	0.73	0.47	0.001	0.004
Cobalamin (mg)	2.14	0.94	2.23	0.88	0.002	0.2
Thiamin (mg)	0.88	0.20	0.86	0.17	0.003	0.2
Sodium (mg)	1370.50	473.38	1331.06	555.28	0.023	0.1
Vitamin E (mg)	0.20	0.09	0.21	0.09	0.059	0.1
Iron (mg)	6.78	1.64	6.84	1.42	0.234	0.01
Selenium (mg)	60.81	12.41	60.39	10.75	0.275	0.01
Niacin (mg)	11.92	2.74	11.84	2.45	0.376	0.01
Riboflavin (mg)	0.82	0.21	0.82	0.20	0.515	0.01
Food groups *						
Fruit (g)	31.24	35.14	50.34	44.62	0.001	5
Fiber (g)	7.66	2.48	8.74	2.59	0.001	4
Vegetables (g)	53.34	33.62	63.39	32.37	0.001	2
Processed Red Meat (g)	11.89	9.09	9.74	7.84	0.001	2
Sugar-Sweetened Beverages (g)	380.57	254.48	329.68	185.59	0.001	1
Read Meat (g)	35.93	19.40	32.62	17.58	0.001	0.8
Dairy (g)	46.38	49.74	54.39	51.95	0.001	0.6
Nuts and Seeds (g)	0.71	2.37	1.29	4.61	0.001	0.6
Wholegrains (g)	4.29	7.93	5.43	9.03	0.001	0.4
Beans and Legumes (g)	18.20	19.14	20.17	17.82	0.001	0.3
Fish (g)	10.08	11.40	10.43	10.82	0.340	0.01

DDS < 5: diverse diet. DDS ≥ 5: non-diverse diet. SD: Standard deviation. η^2^: Eta squared coefficients for estimating the effect size. * Adjusted for 1000 kcal/day.

**Table 4 nutrients-12-01994-t004:** Nutritional status and anthropometric measurements in diet diversity subgroups in ELANS.

Variables	DDS < 5		DDS ≥ 5			
	Mean	SD	Mean	SD	*p* ^1^	η^2^ (%)
Body Weight (kg)	67.32	15.46	66.69	15.26	0.217	0.0
BMI	27.02	5.89	26.88	5.76	0.464	0.0
Waist Circumference (cm)	85.56	14.34	85.27	13.69	0.532	0.0
Neck Circumference (cm)	33.43	3.39	33.49	3.24	0.583	0.0
Hip Circumference (cm)	101.62	12.08	100.95	11.63	0.090	0.0

^1^ Between-group comparisons were adjusted by age, as it was found to be the most influencing factor on nutritional status and anthropometric measurements in relation to the level of DDS (Diet Diversity Score). SD: Standard deviation η^2^: Eta squared coefficients for estimating the effect size.

**Table 5 nutrients-12-01994-t005:** Mean nutrient adequacy ratio of specific nutrients in ELANS.

Nutrient	Overall		DDS < 5		DDS ≥ 5		*p*	η^2^ (%)	*r* ^1^	*p*
	Mean	SD	Mean	SD	Mean	SD				
Magnesium	0.788	0.18	0.725	0.19	0.836	0.16	0.001	9	0.361	0.001
Vitamin A	0.879	0.19	0.813	0.22	0.928	0.14	0.001	9	0.352	0.001
Vitamin C	0.873	0.21	0.799	0.25	0.928	0.15	0.001	9	0.334	0.001
Pyridoxine	0.973	0.08	0.952	0.11	0.989	0.05	0.001	5	0.273	0.001
Vitamin D	0.342	0.20	0.292	0.18	0.380	0.21	0.001	4	0.269	0.001
Cooper	0.982	0.07	0.968	0.09	0.993	0.04	0.001	3	0.216	0.001
Zinc	0.976	0.78	0.960	0.10	0.988	0.05	0.001	3	0.204	0.001
Calcium	0.684	0.45	0.594	0.49	0.750	0.43	0.001	3	0.201	0.001
Phosphorus	0.977	0.08	0.962	0.10	0.989	0.05	0.001	3	0.194	0.001
Folate	0.702	0.17	0.736	0.19	0.677	0.17	0.001	3	−0.205	0.001
Riboflavin	0.987	0.06	0.978	0.08	0.994	0.04	0.001	2	0.183	0.001
Cobalamin	0.980	0.86	0.968	0.11	0.990	0.06	0.001	2	0.174	0.001
Iron	0.976	0.08	0.964	0.10	0.986	0.06	0.001	2	0.162	0.001
Vitamin E	0.031	0.02	0.029	0.02	0.033	0.02	0.001	2	0.148	0.001
Thiamin	0.989	0.06	0.982	0.07	0.994	0.04	0.001	1	0.142	0.001
Niacin	0.996	0.03	0.993	0.04	0.999	0.02	0.001	0.7	0.115	0.001
Selenium	0.998	0.02	0.998	0.02	0.999	0.02	0.135	0.1	0.047	0.004
MAR	0.806	0.06	0.807	0.07	0.850	0.05	0.0001	11	0.393	0.0001

NAR: nutrient adequacy ratio. MAR: The mean adequacy ratio. DDS < 5: non-diverse diet. DD ≥ 5: diverse diet. SD: Standard deviation. η^2^: Eta squared coefficients for estimating the effect size. ^1^ Pearson correlation coefficients (*r*) were calculated between each NAR values and the DDS for the whole sample.

**Table 6 nutrients-12-01994-t006:** Binomial logistic regression model between NAR values and the diverse diet group in ELANS.

Predictor ^1^	Estimate	SE	Z	*p*	Odds Ratio ^2^	95% Confidence Interval	
						Lower	Upper
Intercept	−3.689	0.131	−28.26	0.001	0.03	0.02	0.03
Vitamin C	1.589	0.115	13.78	0.001	4.9 (7.1) ^3^	3.91	6.14
Vitamin A	1.99	0.131	15.21	0.001	7.3 (6.3)3	5.66	9.45
Magnesium	0.962	0.158	6.08	0.001	2.6 (5.0) ^3^	1.92	3.57
Vitamin D	0.849	0.133	6.37	0.001	2.3 (3.4) ^3^	1.8	3.03

^1^ Predictors correspond to NAR (nutrient adequacy ratio) values ^2^ Estimates represent the log odds for belonging to the diverse diet group. ^3^ Odds ratio estimations after adjusting by age, country, and SES. SE: Standard Error.

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
