# Peer review of "Dietary Diversity and Micronutrients Adequacy in Women of Childbearing Age: Results from ELANS Study"

_nutrients, 2020, doi:10.3390/nu12071994_

Round 1

Reviewer 1 Report

I think that the manuscript is very interesting, innovative and well designed, focusing on dietary diversity and micronutrients adequacy in women of childbearing age. The authors recruited a large sample size.

My major concerns are:

  • One limitation of this study is that dietary intake was collected using 24-hours recalls in two non-consecutive days. This method may lead to mistakes about self-reported foods and portions. The authors have to explain exactly how they calculated micronutrients intake.
  • Authors should assess cardiovascular disease (hypertension, diabetes, hypercholesterolemia) in both nondiverse or diverse diets groups of women. There might be a possible relationship between DDS and cardiovascular disease.
  • I suggest to stratify and adjust for Countries the comparisons reported in Table 4 in order to find differences among races and ethnicity (such as waist and hip circumference)
  • Physical activity has not be reported, it may be interesting to include it in the analysis.
  • Drugs addiction and alcohol addiction should be exclusion criteria. The authors included addicted women?

Minor concerns:

1) Line 214: No main effects for the nutritional was observed. I don’t understand this sentence.

Author Response

We will like to thank you for your insightful observations and corrections. Undoubtedly the quality of the manuscript has improved after your carefully revision. Attached you will find the reply to your comments.

Reviewer 2 Report

The content of this publication titled ”Dietary diversity and micronutrients adequacy in women of childbearing age: Results from ELANS STUDY” considers an important problem in modern medical care concerned higher physiological demands related to reproductive role of women of childbearing age. The research concept is well thought-out and carefully planned. The research material and methods were selected correctly in terms of methodology. The study was approved by the local bioethics committee. The results are presented in a clear way. Tables and figures are described correctly, observing all the guidelines used in scientific studies.

The researchers aimed to assess the association between minimum dietary diversity for women and nutrients adequacy ratio, which represents the ratio of a subject’s nutrients intake to the estimated average requirement among  women in childbearing age.

The study was conducted using a large sample of women but there is a pity that data were used from only one 24h recall (instead of minimum 2 or 3). In spite of this limitations, I consider that this work will be satisfied to the readers because of significance of content, scientific soundness.

The Authors show that dietary diversity is limited among women in Latin American countries in childbearing age, so the public interventions in countries from this region are needed to promote an adequate diversity of the diet and knowledge in this field. 

Author Response

Thank you very much, we appreciate your comments. As a matter of fact, we used data from two non consecutive 24-h recalls to compare the consumption of nutrients and food groups between those with a diverse and a non-diverse diet. But, since FAO methodology to assess dietary diversity is intended to be a simple approach, it propose the use of only on 24-h recall, so to estimate dietary diversity we followed they proposal.

Reviewer 3 Report

This study aimed to assess the association between minimum dietary diversity (MDD)  and macro- and micro-nutrient adequacy  among women of childbearing age in Latin America.  The authors used 24-hr recall data  to calculate dietary diversity scores and to assess intakes.  In total, 57.7% of women achieved MDD-W, and women with a more diverse diet had higher micronutrient intakes and healthier intakes in general.

Both major and minor comments are listed here.

1. The authors indicate that the study included women of childbearing age (15-45 years), but there is no reporting of the mean or median age of participants in each country. Why did the authors not control for age in their statistical analyses?   Could differences in mean/median age contribute to the variation in intakes found between Latin American countries?

2. Please elaborate on the choice to use only the first 24 hr recall data, when two days of data were available.  For the data used in the current study, were all data collected on a weekday or weekend?  Having both weekday and weekend data in the current dataset would be problematic, especially given the inclusion of mainly urban populations.

3. Figure 1 - include units on the y-axis.

4. Table 5 - use 'SD', not 'DS' in the column heading.

5. This is a minor point, but please be consistent with the number of micronutrients in each section of the paper.  For instance, in section 2.6, 'NAR was calculated for the 18 micronutrients', but there are only 17 listed (missing sodium).  Later, in line 272, '17 nutrients'.  Then on line 283, - '16 nutrients', although there are 17 in Table 5.  Why was sodium included in Table 3, but not Table 5?

Author Response

Thank you for your insightful observations and corrections. Undoubtedly the quality of the manuscript has improved considerably after your carefully revision. Attached you will find the reply to your comments. 

Reviewer 4 Report

This is an interesting study investigating the association between MDD-W and nutrients adequacy among WCA of eight Latin American countries.

Overall, I think the manuscript was well written and the data was presented clearly. There are a few things I think need to be revised and I have listed them below.

Abstract, remove all statistics, describe the data.

The discussion has a lot of results in it, these should be removed and put in the results section. In the discussion, the authors need to cover the bigger picture as well as discuss how their study contributes to the literature. In the discussion, I think the authors need to comment on the long-term consequences of lower DDS on maternal and offspring health. 

Author Response

Thank you very much for you insightful observations and corrections. Undoubtedly the quality of the manuscript has improved considerably after your carefully revision. Attached you will the find the reply to your comments 

Round 2

Reviewer 1 Report

The paper is well written and the authors explained correctly all my concerns